# Differences in Responses of Immunosuppressed Kidney Transplant Patients to Moderna mRNA-1273 versus Pfizer-BioNTech

**DOI:** 10.3390/vaccines12010091

**Published:** 2024-01-17

**Authors:** Dulat Bekbolsynov, Andrew Waack, Camryn Buskey, Shalmali Bhadkamkar, Keegan Rengel, Winnifer Petersen, Mary Lee Brown, Tanaya Sparkle, Dinkar Kaw, Fayeq Jeelani Syed, Saurabh Chattopadhyay, Ritu Chakravarti, Sadik Khuder, Beata Mierzejewska, Michael Rees, Stanislaw Stepkowski

**Affiliations:** 1Department of Medical Microbiology and Immunology, University of Toledo, Toledo, OH 43614, USA; dulat.bekbolsynov@utoledo.edu (D.B.); andrew.waack@rockets.utoledo.edu (A.W.); camryn.buskey@rockets.utoledo.edu (C.B.); shalmali.bhadkamkar@rockets.utoledo.edu (S.B.); keegan.rengel@rockets.utoledo.edu (K.R.); winnifer.petersen@rockets.utoledo.edu (W.P.); saurabh.chattopadhyay@uky.edu (S.C.); beata.mierzejewska@rockets.utoledo.edu (B.M.); michael.rees2@utoledo.edu (M.R.); 2Department of Urology, University of Toledo, Toledo, OH 43614, USA; mary.brown2@utoledo.edu; 3Department of Anesthesiology, University of Toledo, Toledo, OH 43614, USA; tanaya.sparkle@utoledo.edu; 4Department of Internal Medicine, University of Toledo, Toledo, OH 43614, USA; dinkar.kaw@utoledo.edu (D.K.); sadik.khuder@utoledo.edu (S.K.); 5Department of Electrical Engineering and Computer Science, University of Toledo, Toledo, OH 43614, USA; syedfayeq.jeelani@rockets.utoledo.edu; 6Department of Physiology, University of Toledo, Toledo, OH 43614, USA; ritu.chakravarti@utoledo.edu

**Keywords:** SARS-CoV-2, kidney transplantation, mRNA vaccine, seropositivity, immunocompromised

## Abstract

Immunosuppressed kidney transplant (KT) recipients produce a weaker response to COVID-19 vaccination than immunocompetent individuals. We tested antiviral IgG response in 99 KT recipients and 66 healthy volunteers who were vaccinated with mRNA-1273 Moderna or BNT162b2 Pfizer-BioNTech vaccines. A subgroup of participants had their peripheral blood leukocytes (PBLs) evaluated for the frequency of T helper 1 (Th1) cells producing IL-2, IFN-γ and/or TNF-α, and IL-10-producing T-regulatory 1 (Tr) cells. Among KT recipients, 45.8% had anti-SARS-CoV-2 IgG compared to 74.1% of healthy volunteers (*p* = 0.009); also, anti-viral IgG levels were lower in recipients than in volunteers (*p* = 0.001). In terms of non-responders (≤2000 U/mL IgG), Moderna’s group had 10.8% and Pfizer-BioNTech’s group had 34.3% of non-responders at 6 months (*p* = 0.023); similarly, 15.7% and 31.3% were non-responders in Moderna and Pfizer-BioNTech groups at 12 months, respectively (*p* = 0.067). There were no non-responders among controls. Healthy volunteers had higher Th1 levels than KT recipients, while Moderna produced a higher Th1 response than Pfizer-BioNTech. In contrast, the Pfizer-BioNTech vaccine induced a higher Tr1 response than the Moderna vaccine (*p* < 0.05); overall, IgG levels correlated with Th1(fT_TNF-α_)/Tr1(fT_IL-10_) ratios. We propose that the higher number of non-responders in the Pfizer-BioNTech group than the Moderna group was caused by a more potent activity of regulatory Tr1 cells in KT recipients vaccinated with the Pfizer-BioNTech vaccine.

## 1. Introduction

Severe acute respiratory syndrome coronavirus (SARS-CoV-2) infection has caused significant morbidity and mortality worldwide [1]. The SARS-CoV-2 outbreak rapidly spread coronavirus disease 2019 (COVID-19) with a lower respiratory syndrome as a severe and often deadly pneumonia, especially dangerous in older patients and/or patients with secondary health problems (reviewed in [2]). Most young and healthy people were affected by less severe symptoms of fever, chills, sore throat, myalgia, headache, and anosmia or ageusia [2]. Meanwhile, SARS-CoV-2 virus has continuously mutated with deleterious effects for mutants, as most mutants were swiftly purged [3]. While some “non-purged” mutants remained biologically neutral, a small fraction of mutants transformed COVID-19 outcomes [3]. Over the last two years, the dangerous “beta” SARS-CoV-2 virus mutated into a less deadly but more infectious “omicron” SARS-CoV-2 mutant, causing cold-like symptoms without pneumonia (reviewed in [4]). The most recent FDA recommendation (September 2023) revealed that omicron variant XBB1.5 accounted for 40% of COVID-19 infections in the United States, and thus, steps are being undertaken to produce a boosting dose. Since, the COVID-19 pathogenesis was dependent on the effective virus clearance by the immune system, maintaining vaccine-based protection is crucial. The balance between the viral elimination after vaccination and the control over immune tissue injuries reflects the COVID-19 severity.

SARS-CoV-2 is a single-stranded sense RNA virus with an envelope [5]. The virion includes four membrane proteins, namely, spike (S)1, S2, receptor-binding domain (RBD), and nucleocapsid (N) proteins [6]. Viral entry into cells is mediated by S1 binding to the angiotensin-converting enzyme 2 (ACE2) receptors on cell membranes. This S1/ACE2 association induces a conformational change causing a cleavage of the S2 subunit of transmembrane protease serine 2 (TMPRSS2) enzymes on their cell membranes. The S2 secures a viral/membrane fusion, leading to viral entry into the cell [6]. Blocking S1/ACE2 association and S2/TMPRSS2 function by IgM/IgG/IgA antibodies have been critical for SARS-CoV-2 infection and COVID-19 development [7].

The COVID-19 pandemic prompted efforts to develop an effective vaccine. Out of multiple offers, two mRNA vaccines have been approved by FDA: the mRNA-1273 by Moderna and BNT162b2 mRNA by Pfizer-BioNTech. Following an intramuscular injection, each mRNA vaccine is translated into an immunogenic protein, evoking an immune response [8]. A two-dose regimen of BNT162b2 demonstrated 95% effectiveness in preventing severe COVID-19 [9]. The BNT162b2 vaccine induced humoral and cellular immune responses [10,11]. Based on successful clinical trials, both mRNA vaccines were approved by the FDA [8]. In September 2023, FDA approved the production of a boosting dose targeting the most recent omicron variants including XBB1.5.

Because of continuous immunosuppression, kidney transplant (KT) recipients have an increased risk of COVID-19 [12,13], including severe symptoms requiring hospitalization or even mechanical ventilation. Williamson et al. identified a hazard ratio for mortality at 6.0 for organ transplant recipients, which was significantly higher than that of the general population [14]. When matched for age, gender, and comorbidities, organ transplant recipients had a statistically significant and higher risk of death and/or needed mechanical ventilation [13]. Risk factors contributing to poor outcomes in transplant patients included immunosuppression, obesity, hypertension, cardiovascular disease, and diabetes mellitus [15]. Severe cases of COVID-19 were reported in transplant recipients who received two standard doses of mRNA vaccine [16]. Organ transplant recipients had 68% (95% CI, 58–77) prevalence of anti-SARS-CoV-2 antibodies four weeks after receiving the third dose of BNT162b2 Pfizer-BioNTech; this was higher than 40% prevalence after the second dose. Furthermore, patients who were seropositive after the second dose significantly increased their titers within one month after the third dose (*p* > 0.001). However, even three mRNA doses did not achieve adequate levels of antibodies, thus requiring even more doses [17]. Since an effective vaccination is needed for these vulnerable patients [18,19], a much better understanding of the immune mechanism with a possible involvement of regulatory cells is needed. Current recommendations advise vaccination prior to transplantation whenever possible [20] and a repeated booster vaccination [21].

Considering their high-risk status, transplant recipients were excluded from clinical trials [9,11]. It was concluded from vaccination programs that the vaccination is safe for KT recipients as for other patients [22,23]. Concerns regarding the possible transplant damage or rejection caused by vaccination has not been substantiated [21]. Generally, it is assumed that mRNA-1273 Moderna and BMT162b2 Pfizer-BioNTech vaccines are safe for KT recipients. Unfortunately, ample data have demonstrated a relatively low immunogenicity of the BNT162b2 Pfizer-BioNTech vaccine in transplant patients with low, or even absent seroconversion [24,25,26,27]. Sattler et al. showed a poor cellular response following the vaccination of transplant recipients [26]. There is little information about the quality of immunization due to different mRNA vaccines in recipients.

Our study investigated the effectiveness of mRNA-1273 Moderna and BNT162b2 Pfizer-BioNTech in KT recipients. We measured IgM/IgG/IgA antibody response as well as the frequency of IL-2-, IFN-γ-, and/or TNF-α-producing T-cells. Our analysis showed that mRNA-1273 Moderna vaccine was superior to BNT162b2 Pfizer-BioNTech in KT recipients.

## 2. Materials and Methods

### 2.1. Study Participants

This study involved volunteers and KT recipients tested at the University of Toledo Transplant Center after vaccination against the SARS-CoV-2, who did not have a record of previous SARS-CoV-2 infection. Participants were recruited on a rolling basis from September 2021 through March 2022. KT recipients were vaccinated after transplantation. There was a total of 99 KT recipients and 66 healthy volunteers who were fully vaccinated with the BNT162b2 (Pfizer-BioNTech, Mainz, Germany) or the mRNA-1273 (Moderna, Cambridge, MA, USA) vaccine. The entire cohort was tested 12 months after the last dose of vaccination, and a subgroup of this cohort, 72 KT recipients, and 27 healthy volunteers were additionally tested 6 months after the last dose. Blood samples were collected when participants consented by signing the informed consent form. The study was approved by the University of Toledo Institutional Review Board (IRB# 300931).

### 2.2. Detection and Quantification of Serum Antibodies

Serum IgG and IgM levels were measured using solid-phase sandwich ELISA assay by Invitrogen (catalog #s BMS2324 and BMS2325, respectively), with detection antibodies targeted against SARS-CoV-2 trimerized S-protein domains S1 and S2 pre-coated on plate. Optical density values were read at 450 nm wavelength promptly after 5 min incubation with TMB, and quantitative IgG levels were inferred based on provided standards. For qualitative comparison, we used two definitions of vaccine response. The first definition of seropositivity or seronegativity was based on the manufacturer’s instructions: based on the ratio of absorbance of the sample to the absorbance of the calibrator control. Unlike the responder definition, which was based on a fixed IgG concentration, the seropositivity definition included intermediate values that were treated as inconclusive and were not included in statistical calculations. The second definition of responder or non-responder was based on detectable IgG concentration of 2000 Units/milliliter (U/mL) threshold. Only assays with R2 value of at least 0.9 (≥0.9), indicating a good calibration curve fit, were used.

A subcohort of participants (*n* = 144) was tested for IgA against the N-protein of SARS-CoV-2 using the Gold Standard Diagnostics IgA ELISA test kit (cat # GSD01-1029RUO) for the verification of natural SARS-CoV-2 infection history (See Appendix A).

To confirm the potency of the detected serum antibodies to neutralize the virus, we used the GenScript SARS-CoV-2 surrogate virus neutralization kit (catalog #L00847-A). The kit was designed to test for the presence of antibodies that block the interaction between the receptor-binding domain of the SARS-CoV-2 glycoprotein and the human ACE2 receptor. The total IgG was measured using the Thermo Fisher ELISA kit (BMS2091, Waltham, MA, USA).

### 2.3. ELISpot Assay

Freshly collected whole blood samples were subjected to density gradient spinning to isolate peripheral blood mononuclear cells. The freshly isolated cells were counted using an automatic cell counter and plated at 50,000 cells per well in manufacturer’s ELISpot plates (Immunospot). Cells were incubated for a designated amount of time with or without stimulation. For assaying TNF-α, IL-2, IL-10, and IL-17 secretion, stimulation with LPS at 5 ng/mL was used. After incubation, plates were stained, and spots read using the Immunospot S6 Entry machine.

### 2.4. Statistical Analyses

Mann–Whitney test was used to evaluate the difference in IgG OD_450_ levels between groups: for groups with <50 participants, single-tailed p-values were reported; and for groups with ≥50 participants, two-tailed *p*-values were reported. To assess the relationship between the immune status (controls vs. KT recipients) or vaccine type (Moderna or Pfizer-BioNTech) with binary outcome (seropositive or IgG concentration greater than 2000 U/mL), Chi-squared test was used when each group’s size was at least 5, and Fisher’s exact test was used for comparisons of one or more groups with a size smaller than 5.

The association between a vaccine type and an immune response was evaluated for Th1 and Tr1 readouts. For Mann–Whitney test, participants were divided by an immune status or a vaccine type as described above for the IgG concentration. The frequency of cytokine-secreting cells in an ELISpot assay (spots per 50,000 PBMCs, or spots/5 × 10^4^) was reported after 24–48 h of LPS (5 ng/mL) or PHA-P (5 µg/mL) stimulation. For the frequency of Th1 and Tr1 cells, LPS stimulation was used for comparison between KT recipients and controls, and PHA-P was used for the comparison of participants vaccinated with Moderna vs. Pfizer-BioNTech.

## 3. Results

### 3.1. Cohorts and Patient Characteristics at 6 and 12 Months

For the analysis, we selected KT recipients and healthy controls who were vaccinated against the SARS-CoV-2. Between September 2021 and April 2022, 72 kidney KT recipients and 27 healthy controls were tested at 6 months (Table 1), whereas 99 KT recipients and 66 controls were tested at 12 months. As required by the protocol, all selected participants were fully immunized with either the BNT162b2 (Pfizer-BioNTech) or the mRNA-1273 (Moderna) vaccine, and they were never infected with the SARS-CoV-2 virus; their data are presented in Table 1 for the 6-month group and Appendix A for the 12-month group. The comparison of all participants or KT recipients grouped by the vaccine type showed no significant differences in their social and clinical characteristics. Groups compared by vaccine types at 6 or 12 months had similar distributions of race, gender, clinical immunosuppression, and other variables, thus allowing us to measure their immune metrics related to Moderna vs. Pfizer-BioNTech mRNA vaccine (Table 1 and Appendix A). KT recipients were vaccinated between 30 days to 10 years after transplantation, with an average equal to 2.5 years and a median equal to 6 months.

### 3.2. Anti-SARS-CoV-2 Seropositivity Rates at 6- and 12-Months Post Vaccination

To exclude the possibility of passive transfer of anti-viral IgG, the total IgG levels were measured and confirmed to be similar in healthy volunteers (2629 ng/mL) and KT recipients (2914 ng/mL; *p* > 0.05). One patient with abnormal IgG level was excluded from further analysis. At 6 months, and independent of the mRNA vaccine type, 74.1% of healthy volunteers had anti-SARS-CoV-2 IgG, which was higher than 45.8% among transplant recipients (Figure 1A; Appendix A; *p* = 0.015). This pattern was repeated at 12 months post vaccination: healthy volunteers vaccinated with either mRNA vaccine had 51.5% anti-SARS-CoV-2 IgG seropositivity rate vs. 40.4% in KT patients in the 12-month group (*p* = 0.041; Figure 1B; Appendix A). Overall, mRNA vaccines were more effective in healthy individuals that in KT patients.

By mRNA vaccine type, Moderna vaccine induced an IgG response in 63.8% of all participants (healthy controls + KT patients) vs. 46.9% for Pfizer-BioNTech (*p* = 0.096; Figure 1C; Appendix A) in patients tested 6 months after vaccination. The trend of better Moderna efficacy nearly disappeared at 12 months: Moderna vaccine had a similar effectiveness (50.6% participants) as the Pfizer-BioNTech vaccine (45.1% participants) (*p* = 0.493; Figure 1D; Appendix A).

When observing only KT recipients, the differences were not statistically significant between the two vaccines: 54.3% were IgG seropositive for the Moderna vaccine compared to 40.0% for the Pfizer-BioNTech vaccine (*p* = 0.231; Figure 1E; Appendix A) at 6 months post vaccination. At 12 months post vaccination, IgG was present in 42.8% of KT recipients after immunization with the Moderna vaccine and 40.4% with the Pfizer-BioNTech vaccine (*p* = 0.809; Figure 1F; Appendix A). In a further comparison, healthy volunteers at 6 months after vaccination with the Moderna vaccine were more frequently seropositive (91.7%) vs. those vaccinated with the Pfizer-BioNTech vaccine (64.3%, NS; Appendix A). At 12 months, healthy controls were somewhat similarly seropositive when vaccinated with the Moderna vaccine (61.8%) vs. the Pfizer-BioNTech vaccine (54.2%, *p* = 0.563; Appendix A). Clinical confounders such as age, gender, race, or time post the latest vaccination were not associated with seropositivity for IgG (Table 2).

To better describe the difference in efficiency between Moderna and Pfizer-BioNTech vaccines, we defined responders vs. non-responders by the minimal IgG level of detection (IgG ≤ 2000 U/mL). As shown in Figure 2, all healthy controls vaccinated with either of the mRNA vaccines displayed positive anti-SARS-CoV-2 IgG. In contrast, KT recipients at 6 months had significantly lesser non-responders in the Moderna group (10.8%) than in the Pfizer-BioNTech group (34.3%, *p* = 0.023; Figure 2). A similar trend remained at 12 months with fewer non-responders for Moderna (15.7%) than the Pfizer-BioNTech group (31.3%, *p* = 0.067; Figure 2). Thus, all healthy controls were responders, while all non-responders were KT recipients, with more resulting from the Pfizer-BioNTech vaccination than the Moderna vaccination.

### 3.3. Quantitative IgG Levels at 6- and 12-Months Post Vaccination

Quantitatively at 6 months, KT recipients (63,975 U/mL) had significantly lower IgG levels after either vaccine compared to controls (103,244 U/mL) (*p* = 0.004; Figure 3A). Quantitative data at 12 months also displayed elevated IgG concentrations in healthy volunteers (76,762 U/mL) compared to those of KT patients (57,972 U/m, *p* = 0.036; Figure 3B).

There was also a significant difference in IgG levels between different vaccine types in all participants at 6 months (*p* = 0.041, Figure 3C), and a similar trend was observed at 12 months (*p* = 0.056, Figure 3D). In solely KT recipients, the average IgG concentration was higher in the Moderna group (86,778 U/mL) vs. the Pfizer group (62,829 U/mL) at 6 months (*p* = 0.047, Figure 3E), while a less pronounced difference was observed at 12 months for Moderna (62,352 U/mL) vs. Pfizer-BioNTech (53,314 U/mL, *p* = 0.251, Figure 3F). The Moderna vaccine was slightly more effective in providing protection to healthy volunteers. Unlike IgG, IgM did not show an association with the immune status or vaccine type.

Interestingly, an exclusion of non-responder sera from comparisons resulted in the leveling of the average IgG concentrations between participants vaccinated with either vaccine. The removal of non-responders at 6 months elevated the IgG concentration in Moderna (110,761 U/mL) and Pfizer-BioNTech (67,277 U/mL) groups (*p* = 0.202; Appendix AA,B). At 12 months, the removal of IgG sera-negative KT patients practically reached an average IgG concentration between that of Moderna (76,180 U/mL) and Pfizer-BioNTech (76,701 U/mL, *p* = 0.474; Appendix AC,D). Thus, the lower efficacy of Pfizer-BioNTech stems from the higher number of non-responder patients.

### 3.4. Neutralizing Antibody Frequency at 6- and 12-Months Post Vaccination

We tested all study participants for the presence of virus-specific neutralizing IgM/IgG/IgA antibodies. At 6 months, Moderna-vaccinated participants had more anti-SARS-CoV-2 neutralizing antibodies (81.3%) in comparison with those of Pfizer-BioNTech-vaccinated participants (59.2%, *p* = 0.018, Figure 3G). The same pattern was repeated at 12 months: Moderna-vaccinated participants were more often positive for neutralizing antibodies (79.5%) than Pfizer-BioNTech-vaccinated participants (66.2%, *p* = 0.056, Figure 3H).

Among KT recipients, the Moderna-vaccinated group (75.0%) was more positive for neutralizing antibodies than the Pfizer-BioNTech group (48.6%, *p* = 0.022). This difference was not observed among healthy volunteers as all except two (vaccinated with the Pfizer-BioNTech vaccine) had detectable neutralizing antibodies.

The difference was further clarified when non-responders (which comprised 11% of participants vaccinated with Moderna and 37% of participants vaccinated with Pfizer-BioNTech, *p* = 0.006) were excluded from the comparison. Among the remaining responders, 81% of the Moderna and 73% of the remaining Pfizer-BioNTech participants had neutralizing antibodies. Our observations suggest that a possible common mechanism may be involved in an increased number of non-responders in the Pfizer-BioNTech group.

### 3.5. T Cell Pro-Inflammatory Response to Vaccination

To better explain vaccination efficacy, we evaluated T cell immune responses. PBMCs were explored by an ELISpot assay measuring frequencies of LPS- or PHA-P-reactivated T cells producing IL-2 (fT_IL-2_), IFN-γ, (fT_IFN-α_), and/or TNF-α, (fT_TNF-α_), representing T helper 1 (Th1) cells. The fT_IL-2_ was 2.4-fold lower in KT recipients (1.4 spots per 50,000 PBMCs. 1.4/5 × 10^4^) vs. fT_IL-2_ in controls (3.3/5 × 10^4^, *p* < 0.001; Figure 4A). The fT_IFN-γ_ was 3.6-fold decreased in KT recipients (21.7/5 × 10^4^) compared to fT_IFN-γ_ in controls (79/5 × 10^4^, *p* = 0.016; Figure 4B). Furthermore, the fT_TNF-γ_ was also 2-fold lower in KT recipients (473.8/5 × 10^4^) than fT_TNF-α_ in controls (945.5/5 × 10^4^, *p* < 0.001; Figure 4C), showing twice the expansion of Th1 cells in controls versus that in KT recipients.

Overall, Moderna benefited from Th1 response in KT recipients as the average fT_IL-2_ (11.3/5 × 104) was 19-fold higher than that of Pfizer-BioNTech (0.6/5 × 10^4^, *p* = 0.009, Figure 4D). The fT_IFN-γ_ was 3.1-fold higher for Moderna (40.4/5 × 10^4^) than fT_IFN-γ_ for the Pfizer-BioNTech vaccine (12.8/5 × 10^4^, *p* = 0.022, Figure 4E). Finally, the fT_TNF-α_ was higher (895/5 × 10^4^) for Moderna compared to the fT_TNF-α_ for the Pfizer-BioNTech vaccine (771.1/5 × 10^4^, *p* = 0.311, Figure 4F). Thus, LPS- or PHA-P-reactivated Th1 responses were consistently more robust in healthy volunteers than in KT recipients, and Moderna vaccination generated a stronger Th1 response for KT recipients than did Pfizer-BioNTech vaccination.

### 3.6. T Cell Regulatory Response to Vaccination

One mechanism affecting more potent humoral and cellular responses is active immune regulation. To explore such a possibility, we measured the frequency of IL-10-producing T regulatory 1 cells (Tr1. fT_IL-10_). In KT recipients and healthy controls, the Moderna vaccine had 3.7-fold lower fT_IL-10_ (16.9/5 × 10^4^) than fT_IL-10_ induced by the Pfizer-BioNTech vaccine (62.8/5 × 10^4^, *p* = 0.016; Figure 5A). These results suggest a regulation by Tr1 cells of Th1 cells, resulting in a lower IgG level in Pfizer-BioNTech-immunized patients. In Chi-squared tests, race, gender, and age of recipients had an impact on the fT_IL-10_ cells (Table 2).

The interdependence comparison indicated that the lower fT_IL-10_ correlated with the higher IgG levels (*p* = 0.047; Figure 5B). To confirm this observation, we plotted fT_TNF-α_ and IgG responses: an increased fT_TNF-α_ correlated with a high IgG production (*p* = 0.102; Figure 5C). Finally, the ratio of fT_TNF-α_/fT_IL-10_ also correlated with the IgG production (Figure 5D). An elevated Th1/Tr1 ratio was an indicator of higher IgG levels in vaccinated patients (*p* < 0.001; Figure 5D). Our analysis showed a reciprocal interaction where Th1 and Tr1 cells influenced the IgG production.

## 4. Discussion

KT recipients are at a high risk of severe course and unfavorable outcomes of COVID-19, as confirmed by their higher hospitalization and mortality rates than those of the general population [12,13]. This highlights the importance of an effective vaccination against SARS-CoV-2 in this unique population, which is immunocompromised. Indeed, the rates of seroconversion in vaccinated KT recipients had been shown to suffer due to immunosuppression [24,26,27,28,29]. In addition, the secretion of cytokines, such as Th1-produced IL-2, also has been shown to be lower in transplant recipients compared to that of the general population [26].

As depicted in our summary Table 3, immunosuppressed KT recipients had reduced anti-SARS-CoV-2 IgG response. We propose that the Th1/Tr1 plasticity regulates anti-SARS-CoV-2 IgG response by influencing the rate of responders/non-responder KT patients during a post-transplant vaccination. It seems that the immunogenic SARS-CoV-2 antigens induce potent Th1 cells with weak Tr1 cells in some individuals. In contrast, other individuals develop reduced Th1 cells because of dominant Tr1 cells. This Th1/Tr1 immune regulation correlated with the higher number of non-responders among KT patients vaccinated with Pfizer-BioNTech vaccine (Table 3).

Recently published studies compared two mRNA vaccines for their efficacy [30,31]. Out of 1647 health care workers that tested negative for SARS-CoV-2 antibodies and were vaccinated with two doses of SARS-CoV-2 mRNA vaccines, 688 received the mRNA-1273 Moderna vaccine and 959 received the BNT162b2 Pfizer-BioNTech vaccine [30]. Higher IgG titers were observed after the Moderna than after the Pfizer-BioNTech vaccination (*p* < 0.001) [30]. Participants who were previously infected with SARS-CoV-2 virus and then vaccinated achieved overall higher IgG titers than uninfected participants (*p* < 0.01), but the Moderna vaccine again produced better titers than the Pfizer-BioNTech vaccine (*p* < 0.001) [30]. In a different study, naïve KT recipients vaccinated with the Moderna vaccine developed IgG seropositivity and had T cell ELISpot positivity in two-third of KT patients [31]. In our study, Moderna vaccine induced IgG seropositivity in 54.3% of KT patients compared to only 45.7% with the Pfizer-BioNTech vaccine (*p* = 0.09; Appendix A). Similarly, 65% of Moderna patients were positive for LPS- or PHA-P-reactivated Th1 cells vs. 36% patients vaccinated with Pfizer-BioNTech.

IgG is essential in COVID-19 defense as it fixes the complement to destroy infected cells and opsonizes viral targets for phagocytosis [32]. The rising viral-specific IgG levels following vaccinations are maintained in the following months through memory B cells, conferring the long-term immunity [33]. Efficacy against SARS-CoV-2 relies on IgG as a neutralizing factor, and therefore, the serum levels of SARS-CoV-2-specific neutralizing IgG antibodies reflect the effectiveness of immunization [32]. Indeed, vaccination against SARS-CoV-2 virus correlated with a strong IgG response in an effective defense against COVID-19 symptoms [34]. Our analysis emphasized the generation of IgG in response to anti-COVID-19 vaccination with the neutralizing function correlating with the presence of S1-, S2-, and RBD-specific IgGs. Our new observation was that Moderna was better than Pfizer-BioNTech in KT patients as it increased the number of anti-SARS-CoV-2 IgG-seropositive KT patients. When analyzed by the vaccine type, Moderna also produced better Th1 response than Pfizer-BioNTech, while Pfizer-BioNTech displayed higher levels of IL-10-producing Tr1 cells than Moderna. The lower Th1/Tr1 ratio reflected both depressed Th1 and IgG responses. It is evident that immunosuppression sways the response to mRNA vaccines by the involvement of the Tr1 regulation of Th1, influencing the efficacy of vaccination.

During infection, IL-10 inhibits the activity of Th1 cells, NK cells, and macrophages [35]. On one hand, Th1 cells are required for the optimal pathogen clearance, but on the other hand, the same Th1 activity contributes to tissue damage. Consequently, the best avenue would be for IL-10 not to impede the pathogen clearance but to ameliorate immunopathology. Similarly, the most effective Th1 response to the mRNA vaccination is necessary to produce an efficient memory response to SARS-CoV-2 infection. The downregulation of Th1 response during immunization by Tr1 cells may impede optimal if not maximal protection against SARS-CoV-2 virus in KT patients. Our results showed that the Pfizer vaccine induced a higher Tr1 activity, and this was reflected by lower IgG levels. Our data demonstrate for the first time that active Tr1 regulation is involved in the efficacy of mRNA vaccination in KT recipients.

IL-10, a pleiotropic cytokine with anti-inflammatory functions, acts as a negative regulator of the immune response. In fact, IL-10, including IL-10 produced by Tr1 cells, has been involved in the anti-inflammatory function in autoimmunity, viral/bacterial infections, and allograft transplantation [36,37,38,39,40,41,42]. Alterations in IL-10 producing Tr1 cells were shown to regulate multiple sclerosis, type 1 diabetes, and long-term allograft survival [43,44,45]. Also, in our study, the Pfizer-BioNTech vaccination induced higher Tr1 levels than Moderna, thus possibly contributing to the Th1 downregulation.

In summary, Moderna and Pfizer-BioNTech vaccines are less effective in inducing anti-SARS-CoV-2 IgG and Th1 responses in immunosuppressed KT recipients than in healthy volunteers. While responders after the Moderna or Pfizer-BioNTech vaccine had similar anti-SARS-CoV-2 IgG levels, the Moderna vaccine showed higher benefits for KT patients, with more responder patients than the Pfizer-BioNTech vaccine. We propose that IL-10-producing Tr1 cells contributed to the lower number of IgG responder KT patients in the Pfizer-BioNTech group. However, a limitation was the fact that KT patients had undergone transplantation, but this population also needs future considerations for COVID-19 prevention. Because of random enrollment to our study, patients were not matched with healthy controls. Finally, ELISpot assay analyzed LPS- or PHA-P-reactivated T cells. Future studies will evaluate regulatory mechanisms involved in KT patients.

## 5. Conclusions

In our study, the measurement of immune response metrics in the KT cohort vaccinated after transplantation vs. the healthy cohort revealed the following observations: (1) seroconversion was lower in KT patients than in controls after any mRNA vaccination; (2) seroconversion was higher in KT patients after the Moderna than the Pfizer-BioNTech vaccine; (3) seropositive KT recipients had similar serum anti-SARS-CoV-2 IgG levels after either mRNA vaccine; (4) KT patients had diminished frequencies of LPS- or PHA-P-reactivated Th1 cells (TNF-ɑ, IFN-ɣ, and/or IL-2) compared to controls; (5) Moderna vaccine induced higher Th1 frequencies compared to the Pfizer-BioNTech vaccine; (6) the Pfizer-BioNTech vaccine induced an increased frequencies of IL-10-producing Tr1 cells than the Moderna vaccine; and (7) Th1/Tr1 ratio influenced the anti-SARS-CoV-2 IgG production.

## Figures and Tables

**Figure 1 vaccines-12-00091-f001:**
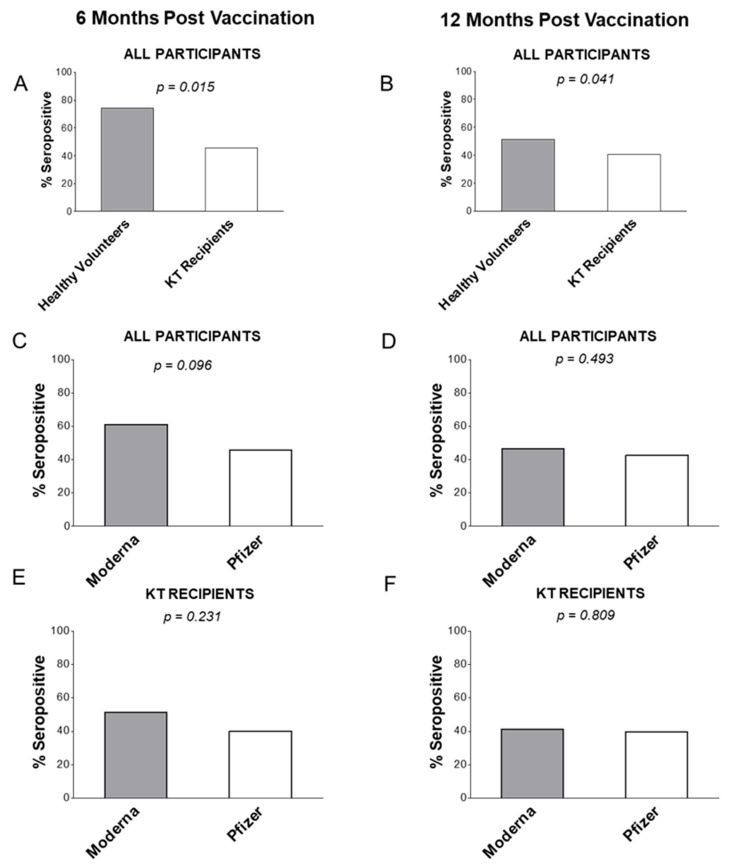
Proportions of IgG-seropositivity results at 6 and 12 months. All participants were divided as healthy controls and KT recipients (**A**,**B**); all participants were divided as those vaccinated with Moderna or Pfizer-BioNTech vaccine (**C**,**D**); and KT recipients were divided as those receiving Moderna or Pfizer-BioNTech vaccine (**E**,**F**). For more details see Materials and Methods.

**Figure 2 vaccines-12-00091-f002:**
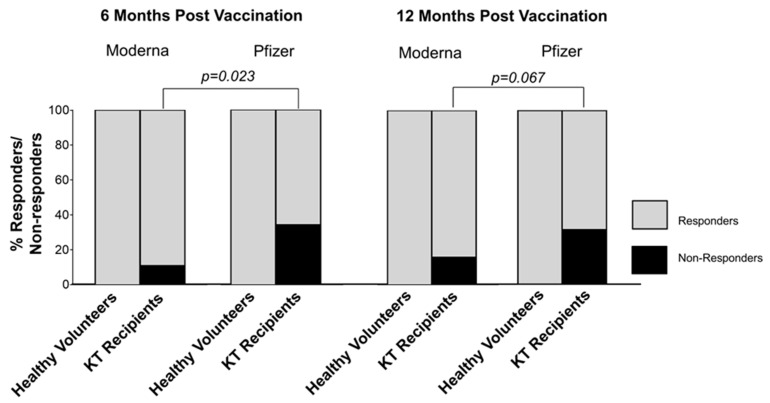
Proportions of responders vs. non-responders to vaccination among healthy controls and KT recipients. Responder or non-responder was defined based on the detectable IgG concentration of 2000 U/mL threshold.

**Figure 3 vaccines-12-00091-f003:**
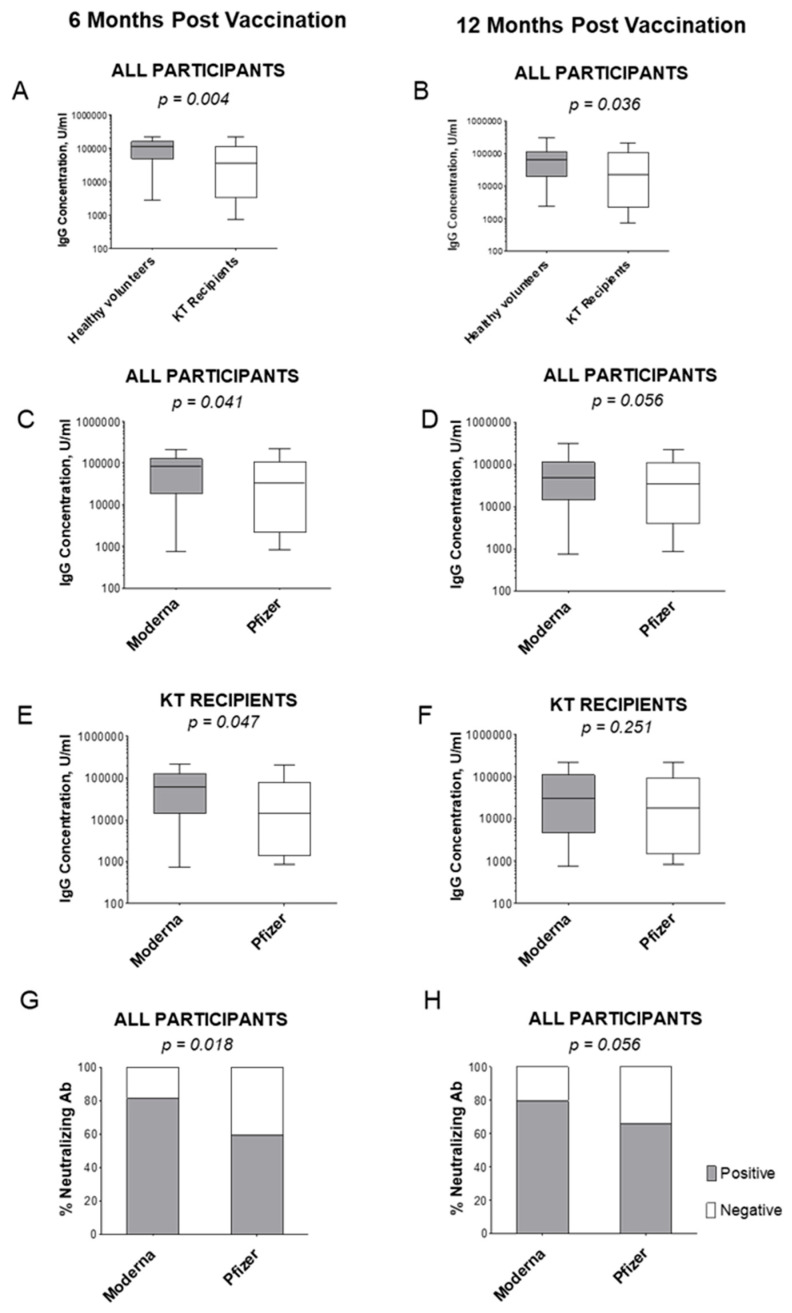
Serum IgG levels and neutralizing antibody frequency. Average serum IgG concentrations are shown in all participants at 6 and 12 months post last vaccination, grouped by cohort (KT recipient or healthy volunteer, (**A**,**B**), and in all participants 6 and 12 months post last vaccination, grouped by vaccine (**C**,**D**), as well as specifically KT recipients grouped by vaccine (**E**,**F**). Percentage of all participants positive for neutralizing antibodies against SARS-CoV-2 was measured in samples at 6 or 12 months post last vaccination (**G**,**H**).

**Figure 4 vaccines-12-00091-f004:**
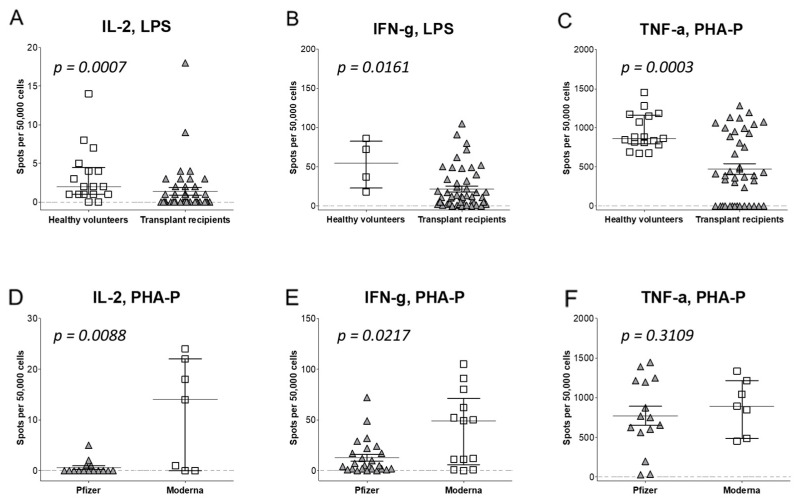
The frequency of LPS- or PHA-P-reactivation of IL-2-(fT_IL-2_; (**A**,**D**)), IFN-γ-(fT_IFN-γ;_ (**B**,**E**)), and TNF-α-(fT_TNF-α;_ (**C**,**F**)), producing T cells in healthy volunteers (squares) compared to KT recipients (triangles) (top panels, (**A**–**C**)), and in KT recipients vaccinated with mRNA-1273 (squares) or Pfizer-BioNTech (triangles) vaccines (bottom panels, (**D**–**F**)), as measured with the ELISpot assay.

**Figure 5 vaccines-12-00091-f005:**
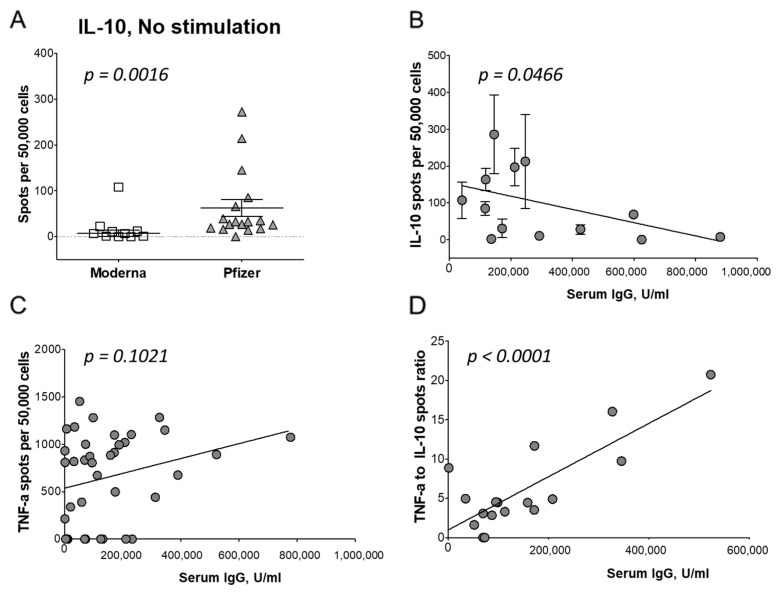
Balance of Th1 and Tr1 response affects the anti-SARS-CoV-2 IgG titer after vaccination. (**A**), KT recipients vaccinated with the mRNA-1273 (squares) produced less spots than PBMCs from KT recipients vaccinated with the Pfizer-BioNTech vaccine (triangles). Correlation between serum IgG levels was shown in KT recipients with IL-10 spots (**B**) and TNF-α spots in ELISpot assay (**C**). Correlation between the ratio of TNF-α to IL-10 spots shown (**D**).

**Table 1 vaccines-12-00091-t001:** Description of participants at 6 months.

Variable	Moderna	Pfizer	*p*-Value ^1^
All Participants (*n* = 98)			
Race			0.048
Caucasian	36 (75.0%)	28 (57.1%)	
Other	12 (25.0%)	22 (44.9%)	
Gender			0.485
Male	27 (56.0%)	32 (44.0%)	
Female	21 (42.9%)	18 (36.0%)	
Age at consent			0.882
35 or less	15 (31.3%)	16 (32.0%)	
36 or more	33 (68.7%)	33 (66.0%)	
Unknown	0 (0.0%)	1 (2.0%)	
Age at consent, years (average)	59.4	56.3	0.177
Vaccine doses 2 + boosting dose			0.598
	18 (37.5%)	22 (44.0%)	
	30 (62.5%)	28 (56.0%)	
Healthy Volunteers (*n* = 27)			
Race			0.276
Caucasian	8 (50.0%)	8 (50.0%)	
Other	4 (36.4%)	7 (63.6%)	
Gender			0.484
Male	4 (36.4%)	7 (63.6%)	
Female	8 (50.0%)	8 (50.0%)	
Age at consent			0.431
50 or less		8 (61.5%)	
51 or more	5 (38.5%)	6 (46.1%)	
Unknown	7 (53.9%)	1 (100.0%)	
Vaccine doses 2 + boosting dose			0.986
	4 (44.4%)	5 (55.6%)	
	8 (44.4%)	10 (55.6%)	
Transplant Recipients (*n* = 71)			
Race			0.006
Caucasian	28 (77.7%)	20 (57.1%)	
Other	8 (22.3%)	15 (42.9%)	
Race			0.351
African American	6 (16.6%)	9 (25.7%)	
Other	30 (83.4%)	26 (74.3%)	
Donor type			0.479
Living	5 (13.9%)	3 (8.6%)	
Deceased	31 (86.1%)	32 (91.4%)	
Gender			0.497
Male	23 (63.9%)	25 (71.4%)	
Female	13 (36.1%)	10 (28.6%)	
BMI			0.465
29 or less	20 (55.6%)	17 (48.6%)	
30 or more	14 (38.9%)	17 (48.6%)	
Unknown	2 (5.5%)	1 (2.8%)	
Age at consent			0.633
35 or less	10 (27.8%)	8 (22.9%)	
36 or more	26 (72.2%)	27 (77.1%)	
Age at consent, years (average)	58.9	58.8	0.490
Vaccine doses 2 + boosting dose			0.411
	14 (38.9%)	17 (48.6%)	
	22 (61.1%)	18 (51.5%)	
Antimetabolites use			0.461
Yes	29 (82.9%)	29 (82.9%)	
No	6 (17.1%)	5 (14.3%)	
Unknown	1 (2.6%)	1 (2.8%)	
CNI use			0.116
Yes	33 (89.2%)	34 (100%)	
No	4 (10.8%)	0 (0%)	
Prednisone use			0.782
Yes	29 (50.6%)	29 (82.9%)	
No	6 (16.7%)	5 (14.3%)	
Unknown	1 (2.7%)	1 (2.8%)	

^1^ Chi-squared test; if in the samples, there were groups of 5 participants or less, Fisher’s exact test was used.

**Table 2 vaccines-12-00091-t002:** Clinical confounders and seropositivity for IgG in the 6-month group.

Group	Positive	Negative ^1^	*p*-Value ^2^
All participants (*n* = 98)			
Gender			0.396
Male	30 (50.8%)	29 (49.2%)	
Female	23 (60.0%)	16 (40.0%)	
Age			0.408
50 years or younger	18 (60.0%)	12 (40.0%)	
51 years or older	35 (50.8%)	33 (49.2%)	
Race			0.600
Caucasian	33 (51.6%)	29 (48.4%)	
Other	20 (55.6%)	16 (44.4%)	
Healthy volunteers (*n* = 27)			
Gender			0.614
Male	9 (81.8%)	2 (18.2%)	
Female	11 (68.8%)	5 (31.2%)	
Age			0.343
50 years or younger	8 (61.5%)	5 (38.5%)	
51 years or older	12 (85.7%)	2 (14.3%)	
Race			0.888
Caucasian	11 (68.8%)	5 (31.2%)	
Other	9 (56.3%)	7 (43.7%)	
Transplant recipients (*n* = 71)			
Gender			0.555
Male	21 (43.7%)	27 (56.3%)	
Female	12 (52.1%)	11 (47.9%)	
Age			0.407
35 years or younger	23 (43.4%)	30 (56.6%)	
36 years or older	10 (55.6%)	8 (44.4%)	
Race			0.458
Caucasian	11 (55.0%)	4 (36.4%)	
Other	9 (45.0%)	7 (63.6%)	
Vaccine doses 2 + boosting dose			0.220
	13 (38.2%)	21(68.2%)	
	20 (54.1%)	17 (45.9%)	

^1^ The “Negative” category includes participants whose ELISA call was intermediate. ^2^ Chi-squared test; if in the samples, there were groups of 5 participants or less, Fisher’s exact test was used.

**Table 3 vaccines-12-00091-t003:** Summary of humoral anti-SARS-CoV-2 IgG as well as cellular Th1 and Tr1 responses in healthy volunteers vs. KT recipients.

Groups	Time Post Vaccination (Months)	SARS-CoV-2 IgG (% Seropositive)		Neutralizing Abs (% Positive)		Th1-IL2/IFNγ/TNFα (Average Spots)		Tr1	
		Moderna	Pfizer	Moderna	Pfizer	Moderna	Pfizer	Moderna	Pfizer
Healthy volunteers	6	91.7	↓60.0 *	100.0	↓30.0 ^	-			
	12	61.8	↓40.4	94.7	92.3	-			
Transplant patients	6	51.4	↓40.0	75.0	↓48.6 ^	IL-2: 11.3 IFNγ: 40.4 TNFα: 895	↓0.6 ^ ↓12.8 ^ ↓771.1	IL10: 24.6	↑78.3 ^
	12	42.9	↓40.4	68.0	↓52.1				
All	6	61.2	↓34.7	81.3	↓59.2 ^	-			
	12	50.6	45.1	79.5	↓66.2	-			

The statistical analysis (*p* < 0.05 *; *p* < 0.001 ^) was performed between Moderna and Pfizer groups; trends were indicated by an arrow down (↓) or an arrow up (↑), showing that the Pfizer group is lower or higher than the Moderna group.

## Data Availability

Unprocessed experimental data provided in Appendix A.

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
