# Peer review of "Differences in Responses of Immunosuppressed Kidney Transplant Patients to Moderna mRNA-1273 versus Pfizer-BioNTech"

_vaccines, 2024, doi:10.3390/vaccines12010091_

Round 1

Reviewer 1 Report

Comments and Suggestions for Authors

This manuscript bring a novel knowlege to the mRNA vaccine immune reaction for kidney transplant patients, it is important for these popuplation. But there is one serious concern need to be addressed:

1. authors claim that moderna effect is better than pfizer, and I admit it is true based on the current studies. However, for the kidy transplant patients, their kidney function is weak, the high IgG, IgA or T cell activity could cause kidney injury. If IgG, IgM, IgA deposit in glomerular area, it can cause high albuminuria. Therefore, authors must provide detailed clinical information about these patients, to tell us their proteinuria, or their kidney function. 

2. authors had better provide complement activation situtation in the serum, because complement system activation, such as C3 can cause kidney injury. 

Author Response

Comment 1:

   …authors claim that moderna effect is better than pfizer, and I admit it is true based on the current studies. However, for the kidy transplant patients, their kidney function is weak, the high IgG, IgA or T cell activity could cause kidney injury. If IgG, IgM, IgA deposit in glomerular area, it can cause high albuminuria. Therefore, authors must provide detailed clinical information about these patients, to tell us their proteinuria, or their kidney function.

Response 1: Thank you for pointing this out. Because creatinine levels were consistently low on average in all our participants, including kidney transplant recipients, we are suggesting that it would be unlikely that immunoglobulin deposition in glomeruli could skew results of our study. To be on the safe side, we also tested total IgG levels in all of our participants and did not find deviations from normal levels.

Comment 2:

authors had better provide complement activation situation in the serum, because complement system activation, such as C3 can cause kidney injury.

Response 2: We appreciate the note that high antibody levels in the Moderna group may have a side effect of immune complex deposition in transplant. While we do not have data on complement regulation or proteinuria in our study participants readily available, we were able to analyze kidney function levels in transplant recipients measured based on patient creatinine and GFR values. We believe that higher antibody levels in Moderna group are unlikely to be associated with impaired kidney function, because creatinine trended lower and GFR was higher in transplant recipients vaccinated with the Moderna vaccine.

Reviewer 2 Report

Comments and Suggestions for Authors

Bekbolsynov et al. investigated the response of kidney transplant (KT) recipients to COVID-19 vaccination. In their study, they enrolled 99 KT recipients and 66 healthy volunteers, administering either the mRNA-1272 Moderna or BNT162b2 Pfizer-BioNTech vaccines to participants. Given that KT recipients are on immunosuppression therapy, their anti-SARS-CoV-2 IgG and Th1 responses are reduced when compared to those of healthy volunteers. Notably, the Pfizer-BioNTech vaccine elicited a stronger T-regulatory 1 (Tr1) response than the Moderna vaccine, which may result in reduced protection for KT recipients. Consequently, the Pfizer-BioNTech KT group showed a higher percentage of non-responders than the Moderna KT group.

Comment:

Kidney recipients undergo immunosuppression to ensure both graft and patient survival. This state of immunosuppression inevitably impacts the efficacy of COVID-19 vaccinations. The significance of this study lies in its implication for the transplant community: among the two vaccines examined, the mRNA-1272 Moderna vaccine may be a more suitable option for immunosuppressed transplant recipients. Overall, the study is well-conducted, and its insights are invaluable for the transplant community. I have no specific concerns.

Author Response

Thank you very much for providing feedback on our work. To seize the opportunity to improve the manuscript in this revision cycle, we made several changes to it, including expanded introduction, verification of history of natural immunization against SARS-CoV-2, exclusion of patients with history of immunoglobulin transfer and improved summary of results in the Discussion section. The changed are reflected in the uploaded version of the manuscript.

Reviewer 3 Report

Comments and Suggestions for Authors

This manuscript presents a study on the immune response in kidney transplant recipients and controls to vaccination with corona vaccine, using Moderna and Pfizer BioNTech mRNA vaccines. The study was conducted in one transplant center. The immune response measured after vaccination included IgM and IgG, detecting antibody activity against trimerized corona viral protein, and included antibodies in a virus-neutralization assay. Cellular immune response in vitro included the frequency of IL-2, Interferon-γ, and/or tumor necrosis factor-α producing T-lymphocytes in peripheral blood. Also the IL-10 and IL-17 response was measured. These responses were processed to the frequency of IL-10-producing T-regulatory 1 (Tr1) cells, and the ratio of Tr1 over Th1 (producing tumor necrosis factor α cells) cells. In total, 66 healthy volunteers and 99 transplant recipients received a full vaccination and were analyzed 12 month after the last vaccination dose: in addition, a subgroup of 27 health volunteers and 12 transplant recipients were tested 6 months after the last dose. A number of parameters were in supplementary information and not available to the reviewer.

The authors summarize the results as follows: 1) seroconversion, i.e. IgG class antibody formation, was lower in transplant patients than in controls after any vaccination; 2) seroconversion was higher in transplant recipients after Moderna vaccination than after Pfizer-BioNTech vaccination; 3) Seropositive transplant recipients had similar serum anti-viral IgG levels after either mRNA vaccine; 4) transplant recipients had diminished frequencies of virus-specific Th1 cells (TNF-α, IFN-γ, and/or IL-2) compared to controls; 5) Moderna vaccination induced higher Th1 frequencies than  Pfizer-BioNTech vaccine; 6) Pfizer-BioNTech vaccination induced an increased frequency of IL-10-producing Tr1 cells than Moderna vaccine; and 7) Th1/Tr1 ratio influenced anti-SARS-CoV-2 IgG production. It is concluded that mRNA Moderna and Pfizer-BioNTech is effective in transplant recipients regarding antibody response as well as the frequency of IL-2-, IFN-γ-, and/or TNF-α-producing T cells. The Moderna vaccine was superior to Pfizer-BioNTech in transplant recipients. But, both vaccines revealed non-responders regarding IgG antibody, which seems to be linked to a change in Tr1/Th1 ratio. Th1 responses were consistently more robust in healthy volunteers than in transplant recipients, and Moderna vaccination generated a stronger Th1 response in transplant recipients than Pfizer-BioNTech vaccination.

This report brings a wealth of new data on the connection between coronavirus vaccination and the reduced immunity status in patients after kidney transplantation. The study design is well described as are the Materials and Methods an presentation of results. The summary of findings is correct, as are the conclusions. The conclusions are correct. The major differences in immune response are highlighted in kidney transplant recipients and non-transplanted volunteers. The combination of assessment of humoral read-outs and cellular parameters is highly interesting and relevant. The relevant difference regarding other studies in this area is that most studies compare vaccination efficacy with the emergence of Covid disease, while this study focuses on immune response differences between transplant recipients and healthy controls.

Comment

·        Following the points mentioned in the foregoing paragraph, the sole focus on immune status after vaccination is at the same time a major point of criticism and a major point regarding study limitations. The authors present the transplant recipients as a rather homogeneous group, which actually might not be the case. See, e.g., Tables 1 and 2. It is mentioned that all transplant recipients are vaccinated after the transplantation. The time after transplantation is not mentioned and it seems logical to assume that the time after transplantation is a major input variable that is highly neglected. Any variation in time after transplantation has a major impact on the immune status, i.e, during induction treatment, maintenance treatment and treatment for rejection. Essentially, there are two aspects in this regard: the immune response to the organ graft and the immune response to the vaccine. The authors only include the immune response to the vaccine. This raises the question whether there was randomization regarding the immune status to the graft. Anyhow, the significant difference between the two vaccinated groups of transplant recipients is rather remarkable in view of the immunosuppressed state related to the transplant status.

·        It is mentioned that the transplant recipients and healthy volunteers received two vaccine doses. It is well known that patients after transplantation have a slow response to COVID vaccins, and that three vaccinations might be needed  to receive the same efficacy as two vaccines in individuals that are not transplanted. See, e.g., Kamar et al, Three Doses of an mRNA Covid-19 Vaccine in Solid-Organ Transplant Recipients. N Engl J Med. 2021 Jun 23 : NEJMc2108861.doi: 10.1056/NEJMc2108861: this study was performed with the Pfizer-BioNTech vaccine which appeared less efficacious than the Moderna vaccine in the present report. This raises the question why in the present study the transplant recipients did not receive three doses, and, if they had received three doses whether the outcomes regarding immune parameters would have been different (less difference between transplant recipients and healthy volunteers). Also, would the use of three dose have had efficacy so that there were no non-reponders?

·        Following the above the demographic data of the transplant patients need to be presented in much more detail, and also, whether transplant patients were matched with healthy volunteers or not (and why not). Also, an important limitation is the absence of immune parameter assessment before transplantation, and before vaccination. Note in this regard  that kidney transplant recipients often had a (long) period of dialysis before transplantation with an inherent depressed immunity status.

·        It is highly advised to present the text in the results in a more readable form, and consider to summarize results in summary tables in the summary paragraph. If numerical are present in tables/figures they do not to be presented in the text. Also throughout the body text output parameter are presented that do nut return in the result like IgG, IgM and IL-7 etc.

·        It is highly advise to include paragraph summarizing the study limitations.

·        It is highly advised to include a paragraph presenting the perspectives of this study.

Author Response

Comment 1:

   …Following the points mentioned in the foregoing paragraph, the sole focus on immune status after vaccination is at the same time a major point of criticism and a major point regarding study limitations. The authors present the transplant recipients as a rather homogeneous group, which actually might not be the case. See, e.g., Tables 1 and 2. It is mentioned that all transplant recipients are vaccinated after the transplantation. The time after transplantation is not mentioned and it seems logical to assume that the time after transplantation is a major input variable that is highly neglected. Any variation in time after transplantation has a major impact on the immune status, i.e, during induction treatment, maintenance treatment and treatment for rejection. Essentially, there are two aspects in this regard: the immune response to the organ graft and the immune response to the vaccine. The authors only include the immune response to the vaccine. This raises the question whether there was randomization regarding the immune status to the graft. Anyhow, the significant difference between the two vaccinated groups of transplant recipients is rather remarkable in view of the immunosuppressed state related to the transplant status.

Response 1: Thank you for pointing this out. The note on cohort heterogeneity was indeed important to address in the manuscript, and we agree that patients included in our study are different in various regards, and more details could have been mentioned in the Methods section about our participant recruitment process. We therefore expanded on our description of patients’ recruitment in Section 2.1 of the manuscript (see attached). That being said, we selected our patient cohorts in such fashion that they would appear statistically similar (with Chi-squared test yielding high p-values) in the key demographics listed in Tables 1 and 2. Those characteristics are: gender, age, race as well as the maintenance immunosuppression. What did slip out of our attention is the time between the transplantation time and the involvement into our study. What is consistent in our study is the fact that all our transplant patients had been vaccinated after kidney transplantation when all are undergoing immunosuppression.

Speaking about the time post transplantation, we have transplant date data readily available for 48 study participants within the 12 months post last vaccination period. The time post transplantation to consent for participation in the study ranged from 2 days to 10 years post-transplantation, with an average time being equal to 2,5 years and a median equal to 6 months. Levels of IgG and rates of seropositivity did not appear to correlate with the time post transplantation, although an additional controlled study would be needed to make a definitive conclusion. For clarification, we included an additional description of these data in the updated version of the updated manuscript.

Comment 2:

It is mentioned that the transplant recipients and healthy volunteers received two vaccine doses. It is well known that patients after transplantation have a slow response to COVID vaccins, and that three vaccinations might be needed  to receive the same efficacy as two vaccines in individuals that are not transplanted. See, e.g., Kamar et al, Three Doses of an mRNA Covid-19 Vaccine in Solid-Organ Transplant Recipients. N Engl J Med. 2021 Jun 23 : NEJMc2108861.doi: 10.1056/NEJMc2108861: this study was performed with the Pfizer-BioNTech vaccine which appeared less efficacious than the Moderna vaccine in the present report. This raises the question why in the present study the transplant recipients did not receive three doses, and, if they had received three doses whether the outcomes regarding immune parameters would have been different (less difference between transplant recipients and healthy volunteers). Also, would the use of three dose have had efficacy so that there were no non-reponders?

Response 2: We agree with the comment. It can also be mentioned that most of the patients for this study were recruited in September-December of 2021. During that period in the United States, the Moderna and Pfizer vaccines had been recently approved for prevention of COVID-19 disease. There was consensus among healthcare practitioners that healthy individuals should be vaccinated with two doses of these mRNA-vaccines, however, transplant recipients had started participation in vaccination at later time, and there was no sufficient number of patients vaccinated before transplantation. As suggested, we have included an extensively updated explanation with relevant recent references in the following text: “Severe cases of COVID-19 were reported in transplant recipients who received standard two doses of mRNA vaccine. Organ transplant recipients had 68% (95% CI, 58-77%) prevalence of anti-SARS-CoV-2 antibodies four weeks after receiving the third dose of BNT162b2 PfizerBioNTech; this was higher than 40% prevalence after the second dose. Furthermore, patients who were seropositive after the second dose significantly increased their titles within one month after the third dose (p>0.001). However, even three mRNA doses did not achieve adequate levels of antibodies, thus requiring even more doses. Since an effective vaccination is needed for these vulnerable patients much better understanding is needed of the immune mechanism with a possible involvement of regulatory cells. Current recommendations advise vaccination prior to transplantation whenever possible and a repeated booster vaccination.”

Comment 3:

Following the above the demographic data of the transplant patients need to be presented in much more detail, and also, whether transplant patients were matched with healthy volunteers or not (and why not). Also, an important limitation is the absence of immune parameter assessment before transplantation, and before vaccination. Note in this regard  that kidney transplant recipients often had a (long) period of dialysis before transplantation with an inherent depressed immunity status.

Response 3: We agree that this study has been performed as a cross-sectional assessment of immune response metrics in patients already vaccinated against the virus and this may be considered a limitation of the design, as well as the fact that healthy volunteers in this study have not been matched to kidney transplant patients.

Comment 4:

It is highly advised to present the text in the results in a more readable form, and consider to summarize results in summary tables in the summary paragraph. If numerical are present in tables/figures they do not to be presented in the text. Also throughout the body text output parameter are presented that do nut return in the result like IgG, IgM and IL-7 etc.

  • It is highly advise to include paragraph summarizing the study limitations.

  • It is highly advised to include a paragraph presenting the perspectives of this study.

Response 4: Thank you for this comment. We agree that this manuscript will benefit from improving clarity and readability of the results. We performed the following modifications in the manuscript text:

  • A table summarizing the findings of immune response metrics has been added to the Discussion section.
  • The Results section has been edited to reduce the clutter of numbers.
  • Also, a paragraph summarizing the main findings of the study has been added in the Discussion section.
  • As requested we added the following text about limitations and future studies “While the limitation was the fact that KT patients were after transplantation, this population also needs future considerations for COVID-19 prevention. Because of random enrollment to our study, patients were not matched with healthy controls. Finally, ELISPOT assay analyzed LPS- or PHA-P-reactivated T cells. Future studies will evaluate regulatory mechanisms involved in KT patients”.

Reviewer 4 Report

Comments and Suggestions for Authors

Bekbolsynov et al. reported in their study differences in the response of immunosuppressed kidney transplant recipients to various mRNA vaccines. Although the study covers very important topic, there are several critical limitations in methodology, study design and data interpretation:

1) There is a different number of patients receiving booster vaccine doses at month 6 (8 and 10) and month 12 (20 and 17), suggesting different times of booster vaccination.

2) Although only patients without any record of previous SRS-CoV-2 infection were enrolled, it is important to assess anti-SARS-CoV-2 specific antibodies before treatment since asymptomatic infection may occur. Different responses can be expected in patients with hybrid immunity compared to vaccinated-only individuals.

3) The authors did not provide any data on total immunoglobulin levels. Did any patients receive immunoglobulin replacement therapy - IRT? I would expect secondary antibody immunodeficiency in the number of patients with immunosuppression. In the case of IRT, the results can be skewed by passively transmitted antibodies.

4) Did the authors detect antigens against RBD or any other antigens? It is not specified in the methods.

5) The authors assessed T-cell response after LPS stimulation in the context of vaccine efficacy. Nevertheless, this assay cannot detect SARS-CoV-2-specific T-cells.

6) The authors provided no data on cellular immunity, such as lymphocyte population/subpopulation counts, including CD3, CD4, and CD8 cells. Similarly to humoral response, I would expect secondary T-cell immunodeficiency in the number of patients with immunosuppression, which may also significantly affect cellular response. 

Comments on the Quality of English Language

Minor editing of English language required

Author Response

Comments 1: There is a different number of patients receiving booster vaccine doses at month 6 (8 and 10) and month 12 (20 and 17), suggesting different times of booster vaccination.

Response 1: We appreciate you bringing up this important point about vaccination times. We agree that more clarification is needed for the description of study participants recruitment in the Methods section of the manuscript. Indeed, the necessary modifications have been made in the manuscript (see attached). Challenging was the fact that we had been recruiting patients into the study on the ongoing basis from August 2021 to March 2022. Consequently, the participants included in the 12-months cohort were recruited at different times within this window, which then explains why the number of people who received a booster vaccine dose is not the same as in the 6-months cohort.

Comments 2: Although only patients without any record of previous SRS-CoV-2 infection were enrolled, it is important to assess anti-SARS-CoV-2 specific antibodies before treatment since asymptomatic infection may occur. Different responses can be expected in patients with hybrid immunity compared to vaccinated-only individuals.

Response 2: Thank you for bringing up this important point. We agree, the description of Methods in the manuscript should have included more details on the recruitment of study participants. While we included transplant patients in the study, based--among other things---on self-reported absence of previous coronavirus disease infections. However, in addition we did perform an ELISA test for the IgA antibidies against the nucleocapsid protein of SARS-CoV-2 (N-antigen), which is a known indicator of the natural SARS-CoV-2 infection. The total number of 104 out of 165 total participants were tested and 18 patients were positive for N-antigen specific IgA, thus confirming COVID-19 infection. We included these new results in the Supplemental Data section of the updated manuscript (see attached).

Comments 3: The authors did not provide any data on total immunoglobulin levels. Did any patients receive immunoglobulin replacement therapy – IRT? I would expect secondary antibody immunodeficiency in the number of patients with immunosuppression. In the case of IRT, the results can be skewed by passively transmitted antibodies.

Response 3: Thank you for this important comment. As requested, we performed an additional total IgG test to address this concern. The total IgG level was measured by the Thermo Fisher ELISA kit (BMS2091). The following text has been added in the Results section “To exclude the possibility of passive transfer of anti-viral IgG, total IgG levels were measured and confirmed to be similar in healthy volunteers (2,629 ng/ml) and KT recipients (2,914 ng/ml; NS; not shown). One patient with abnormal IgG level was excluded in further analysis.”

Comments 4: Did the authors detect antigens against RBD or any other antigens? It is not specified in the methods.

Response 4: We appreciate you catching this missing part of the Methods section. As suggested, we modified the Methods section accordingly. In brief, the tests for IgG and IgM were designed to detect antibodies specific to the S1/S2 antigen of the spike protein of SARS-CoV-2. The tests for IgA antibodies were aimed at detecting antibodies specific to domains S1/S2/RBD and the Nucleocapsid protein (N-antigen).

Comments 5: The authors assessed T-cell response after LPS stimulation in the context of vaccine efficacy. Nevertheless, this assay cannot detect SARS-CoV-2-specific T-cells.

Response 5: Thank you for pointing out these important issues. Accordingly, we have explicitly explained that “PBMCs were explored by an ELISpot assay measuring frequencies of LPS- or PHA-P-reactivated T cells producing IL-2 (fTIL-2), IFN-γ, (fTIFN-α), and/or TNF-α, (fTTNF-α), representing T helper 1 (Th1) cells.”

Comments 6: The authors provided no data on cellular immunity, such as lymphocyte population/subpopulation counts, including CD3, CD4, and CD8 cells. Similarly to humoral response, I would expect secondary T-cell immunodeficiency in the number of patients with immunosuppression, which may also significantly affect cellular response.

Response 6: Thank you for this comment. We plan to perform more detailed immunophenotyping the PBMCs collected from transplant patients and healthy volunteers. One of the limitation of our study was the IRB advisory committee limiting the amount of blood within this study justified by the research goal. 

Round 2

Reviewer 1 Report

Comments and Suggestions for Authors

Great, much improved, and I recommend it for publication. 

Reviewer 3 Report

Comments and Suggestions for Authors

The authors have carefully considered the comments made by the reviewers. The (detailed) responses to the comments are appreciates. Also a number of text edits have been made.